# *SpPKE1,* a Multiple Stress-Responsive Gene Confers Salt Tolerance in Tomato and Tobacco

**DOI:** 10.3390/ijms20102478

**Published:** 2019-05-20

**Authors:** Jinhua Li, Chunrui Chen, Juanjuan Wei, Yu Pan, Chenggang Su, Xingguo Zhang

**Affiliations:** 1State Cultivation Base of Crop Stress Biology for Southern Mountainous land of Southwest University, Academy of Agricultural Sciences, Southwest University, Beibei, Chongqing 400715, China; ljh502@swu.edu.cn (J.L.); eline1020@email.swu.edu.cn (C.C.); shucaitomato@gmail.com (J.W.); pany1020@swu.edu.cn (Y.P.); suchenggang@swu.edu.cn (C.S.); 2Key Laboratory of Horticulture Science for Southern Mountainous Regions, Ministry of Education, College of Horticulture and Landscape Architecture, Southwest University, No.2 Tiansheng Road, Beibei, Chongqing 400715, China

**Keywords:** abiotic stress, salt stress, proline-, lysine-, and glutamic-rich protein, wild tomato species, tobacco

## Abstract

Understanding the mechanism of abiotic-tolerance and producing germplasm of abiotic tolerance are important in plant research. Wild species often show more tolerance of environmental stress factors than their cultivated counterparts. Genes from wild species show potential abilities to improve abiotic resistance in cultivated species. Here, a tomato proline-, lysine-, and glutamic-rich type gene *SpPKE1* was isolated from abiotic-resistant species (*Solanum pennellii* LA0716) for over-expression in tomato and tobacco for salt tolerance. The protein encoded by *SpPKE1* was predominantly localized in the cytoplasm in tobacco. SpPKE1 and SlPKE1 (from cultivated species *S. lycopersicum* cv. M82) shared 89.7% similarity in amino acid sequences and their transcripts abundance in flowers and fruits was reduced by the imposition of drought or oxidative stress and the exogenous supply of abscisic acid. The DNA of the *PKE1* promoter was highly methylated in fruit and leaf, and the methylation of the coding sequence in leaf was significantly higher than that in fruit at different development stages. The over-expression of *SpPKE1* under the control of a CaMV (Cauliflower Mosaic Virus) 35S promoter in transgenic tomato and tobacco plants enhanced their tolerance to salt stress. *PKE1* was downregulated by abiotic stresses but enhanced the plant’s salt stress tolerance. Therefore, this gene may be involved in post-transcriptional regulation and may be an important candidate for molecular breeding of salt-tolerant plants.

## 1. Introduction

Plant growth and development are continuously influenced by environmental factors, including water content, salinity, and temperature, which may disrupt plant homeostasis. The tomato plant belongs to the *Solanaceae* family. It is mainly cultivated as edible fruit and is a kind of vegetable crop which can be commercially grown worldwide (FAOSTAT 2013). Production and consumption of tomato has been increasing continuously [1]. Although some available tomato germplasms possess tolerance for various abiotic stresses, crop losses will become severe under extreme climatic conditions. Domestication, selection, and different breeding methods have narrowed down the genetic diversity of cultivated tomato. Therefore, developing new high-yielding cultivars, that are tolerant to various abiotic stresses, will substantially influence tomato production [2].

Transgenic plants with engineered endogenous genes produce abiotic stress-tolerant phenotypes in tomato. These genes affect abiotic stress tolerance. *S. lycopersicum* ethylene response factor B.3 (*Sl-ERF.B.3*) gene encodes for a tomato ethylene-responsive element binding factor (ERF) transcription factor, whereas *Sl-ERF.B.3* antisense transgenic plants exhibit salt- and cold-stress-dependent growth inhibition [3]. A tomato class II ERF protein SlERF3 also plays an important role in crop improvement or genetic engineering by increasing stress tolerance in plants [4]. Tomato-dehydration-responsive element binding two transcription factors enhances plant tolerance to salinity in tomato and *Arabidopsis* [5]. Similarly, the tomato zinc finger2 cysteine-2/histidine-2 repressor-like transcription factor confers tolerance to salinity in tomato and *Arabidopsis* [6]. A tomato bZIP protein named abscisic acid-responsive element binding protein (AREB1) is responsive to abscisic acid (ABA) and it increases the salt and water loss resistance of the tomato plant [7]. Functional proteins such as Na^+^/H^+^ antiporter SlSOS1 and K^+^/H^+^ antiporter NHX2 are also successfully used to improve tomato salt stress tolerance [8,9,10]. Transgenic tomato plants with overexpressed *AtNHX1* increased the capacity to retain intracellular K^+^ and confer salt stress tolerance [11]. The ectopic expression of tomato *SlTIP2;2* can enhance tolerance to salt stress in *Arabidopsis* [12]. In some cases, genes negatively function in abiotic stress resistance. For example, NAC transcription factor SlSRN1 of tomato is not only a negative regulator for oxidative and drought stress response [13], but also a negative regulator of salt and oxidative stresses as a kind of hybrid proline-rich protein (PRP) [14].

In this study, we isolated a proline-, lysine-, and glutamic-rich protein gene (*PKE*) from drought-resistant species (*S. pennellii* LA0716) and cultivated species (*S. lycopersicum* cv. M82) and named as *SpPKE1* and *SlPKE1* respectively. *PKE1* was differentially expressed after drought treatment of *S. lycopersicum* cv. M82 and *S. pennellii* LA0716 [15]. Then, expression and function of this gene were analyzed. *PKE1* is suppressed by various abiotic stresses, including dehydration, oxidative stress, and phytohormones ABA and salicylic acid (SA). Over-expression (OE) of *SpPKE1* can significantly enhance salt tolerance of tomato and tobacco. DNA of the *PKE1* coding sequence in leaf is methylated higher than that in fruit at different development stages. This research work will shed more light on the molecular mechanism of *PKE1*, which is down-regulated by stress but plays a positive role in stress tolerance.

## 2. Results

### 2.1. Characterization of PKE1 in Tomato

In a previous study of drought stress in tomato introgression lines [15], a differential expression profile of the *PKE1* gene was observed between the drought-tolerant introgression line (IL) and M82. After drought stress, the expression of *PKE1* significantly decreased in M82, IL2-5 and IL9-1 (Appendix A). Full-length *PKE1* cDNAs were isolated from *S. lycopersicum* cv. M82 and *S*. *pennellii* LA0716 by reverse transcription (RT)-PCR and labelled as *SlPKE1* and *SpPKE1* respectively. *SlPKE1* and *SpPKE1* encoded 326 and 319 amino acids respectively, which shared 89.7% similarity (Figure 1A). Proline in SlPKE1 and SpPKE1 accounted for 18.7% and 18.5% of the total amino acid residues respectively, followed by lysine (17.2% and 16.3%) and glutamate (12.3% and 11.2%). The proline, lysine and glutamate acid residues in SlPKE1 and SpPKE1 accounted for 48.2% and 46.1% of the total residues respectively (Appendix A). According to tomato genomic sequence, *SlPKE1* and *SpPKE1* both contain two introns and three exons, but have difference in nucleotide lengths of the third exon (Figure 1B). Hence, *PKE1* is conserved in wild and cultivated plant species and code divergent amino acids. The phylogenetic tree, constructed based on the amino acid sequences of PKE1 and PKE proteins from other representative organisms, demonstrated that SlPKE1 and SpPKE1 were evolutionarily closely related to those PKEs, isolated from other *Solanaceae* plants (*S. tuberosum* and *S. chacoense*) (Figure 1C). These results implied that PKE1 might have similar functions as other PKE proteins, isolated from *Solanaceae* plants.

*Cis*-elements participate in gene regulation by interacting with their corresponding trans-regulatory factors. Hence, the promoter regions of *SpPKE1* and *SlPKE1* were retrieved and submitted to the PlantCARE database for *cis*-element identification (Table 1). Conventional promoter elements (TATA-box and CAAT-box) were detected in *PKE1* promoters. The remaining *cis*-acting elements can be divided into four groups. Six *cis*-elements, Box 4, ARE (only in *SlPKE1*), AE-box, ATCT-motif, LAMP-element (special appeared in *SlPKE1*), and TCT-motif, were light-responsive. Three *cis*-elements, CGTCA-motif, TGACG-motif, and TCA-element (only in *SlPKE1*), were hormone-responsive. Five *cis*-elements, TC-rich repeats, LTR (only in *SpPKE1*), MYB, MYB-like sequence, and MYC, functioned as stress-responsive elements. The fourth group had a HD-Zip 1 *cis*-element, which was involved in the differentiation of palisade mesophyll cells. 

### 2.2. SpPKE1 Localizes to the Cytoplasm

To determine the sub-cellular localization of SpPKE1, we fused the full-length open reading frame of *SpPKE1* with the N-terminal of a green fluorescent protein (GFP) reporter protein driven by CaMV 35S promoter and generated a fusion protein SpPKE1-GFP. The fusion protein was infiltrated into tobacco suspension cells. Microscopic observation demonstrated that green fluorescence in the transformed cell was mainly localized in the cytoplasm, whereas no fluorescence was detected in non-transformed cells. Besides fluorescence was displayed throughout their structures (Figure 2) in those cells, which were transformed with the vector containing only GFP. 

### 2.3. PKE1 Expression Suppression by Abiotic Stress and Hypermethylation in Different Tissues

RT-PCR analysis showed similarity of PKE1 tissue expression between *S. pennellii* and *S. lycopersicum* cv. M82. Indeed, PKE1 was highly expressed in flower and fruit, while a lower expression was observed in root and leaf (Figure 3A,B). DNA methylation analysis indicated that PKE1 promoter and the coding sequences were highly methylated, and accordingly with gene expression, methylation was significantly higher in the leaf compared to flower and fruit (Figure 3C and Appendix A). Interestingly, PKE1 expression was significantly suppressed by various abiotic stresses, including drought, methyl viologen (MV), ABA, and SA (Figure 4). Under drought, MV, GA3, and SA treatments, PKE1 expression was gradually reduced, and *S. pennellii* and *S. lycopersicum* cv. M82 showed similar expression patterns. However, PKE1 expression returned to pre-treatment levels after 24 h from ABA and Eth treatment. Under ABA treatment, PKE1 expression was significantly suppressed after 1 h of treatment in *S. pennellii*, while none variation in gene expression was observed in *S. lycopersicum* cv. M82. By contrast, under Eth treatment, PKE1 expression was significantly suppressed after 1 h of treatment in tomato cv. M82 but did not exhibit response in *S. pennellii*.

### 2.4. PKE1 Overexpression Enhanced Tolerance to Salt Stress

Salt tolerance was examined in *PKE1* overexpression (OE) and RNAi-konckdown (Ri) transgenic plants. The significant *PKE1* OE plants (OE2 and OE5) and knockdown *PKE1*-RNAi plants (Ri1) were selected for further analysis. As shown in Figure 5, the *SpPKE1* OE lines presented significantly enhanced salt tolerance, and the Ri1 was more sensitive compared to wild type (WT) lines. After salt treatment, less leaf necrosis and increasing shoot biomass was observed in transgenic OE plant lines. The survival rate in OEs (OE2: 79% and OE5: 75%) is higher than that in WT (33%) and Ri (8%). The result indicates that tomato transformed with the *SpPKE1* gene displayed salt tolerance.

For further functional analysis of *PKE1,* 31 *PKE1* OE transgenic tobacco plants were generated, and expression level transgenic plants was examined. *PKE1* with significant differential expression compared with wild type (WT) and OE (OE20, OE28 and OE31) plants were selected for further analysis. No morphological difference was noted between the transgenic lines and WT (data not shown). The seedling shifted to the high salt for treatment for 7 days, the WT seedling leaves started to turn yellow, and the transgenic OE plant remained green (Figure 6A). After 12 days of treatment, the length of the root was significantly reduced. However, this reduction was significantly higher in WT lines than in *SpPKE1*-overexpressing plants (Figure 6B). Moreover, the content of chlorophyll was determined. Figure 6C shows that the chlorophyll contents of OE lines were significantly higher than those of WT plants under salt stress. Malondialdehyde (MDA) content was also determined under salt stress in WT and transgenic plants. As it is shown in the Figure 6D, MDA increased in WT and transgenic lines after salt treatment. However, this increase in salt-stressed WT lines was significantly higher than that in *SpPKE1* OE plants. It indicates that the *SpPKE1* OE plant inhibited the increase of MDA content under salt stress. 

Together, our results were strongly supportive of *SpPKE1* OE plants, which confer salt stress tolerance in both tomato and tobacco. 

## 3. Discussion

In our study, major observations regarding *SlPKE1* and *SpPKE1* included their expression in flowers and their negative regulation by ABA and abiotic stresses (Figure 3 and Figure 4). SpPKE1 was a cytoplasm protein (Figure 2), and their OEs as transgenes enhance salt tolerance (Figure 5 and Figure 6). Therefore, *PKE1* was down-regulated by ABA, and drought stress enhanced salt stress tolerance.

### 3.1. PKE1 Gene Sequence Diversity between Solanum Species and Involvement in Abiotic Stress

Domesticating and growing plants as crops can present both advantages and disadvantages. Selecting specific desirable traits, such as high yield, can increase crop productivity, but other important traits, such as abiotic resistance, might be lost. It might induce vulnerability of crops to different stresses. Researchers often use wild relatives of crops to reduce these vulnerabilities. In our study, PKE1 was isolated from drought-resistant (*S. pennellii* LA0716) and cultivated species (*S. lycopersicum* cv. M82). The amino acid sequences of SpPKE1 and SlPKE1 shared high similarity, and the promoter *cis*-element of *SpPKE1* and *SlPKE1* had three differentiations (Figure 1 and Table 1). This finding implies that PKE1 is a protein that shows evolutionary trace between *S. lycopersicum* and *S. pennellii*. 

PKE1 is a proline-, lysine-, and glutamic-rich protein (Figure 1A), and single domain in proteins such as proline-rich and lysine-rich protein have been reported to have important functions on abiotic tolerance [16,17,18]. It suggests that PKE has been involved in plant abiotic response. The proline-rich proteins (PRP) are initially defined as proteins involved in response to wounding [19]. A HyPRP1 gene in tomato plays a negative role in abiotic stress tolerance [14]. Heterologous expression of a lysine-rich protein gene *SBgLR* in potato can significantly increase maize salt resistance [20]. Moreover, both the promoter of *SpPKE1* and *SlPKE1* had an MYB and MYC *cis*-element (Table 1). The MYB and MYC proteins are major transcription factors that play significant roles in plant defenses against various stresses [21,22,23,24,25,26,27,28,29]. These findings, together with our results, show that the MYB or MYC transcription factor may directly regulate PKE1 to confer abiotic stresses.

### 3.2. PKE1 Down-Regulated by Abiotic Stresses and Enhance Salt Tolerance

In most cases, the suppression of negative regulators or enhancement of positive regulators of ABA will confer abiotic tolerance [5,14,30,31,32,33]. However, reports about this topic provide different conclusions. A wheat zinc finger gene TaCHP was downregulated by ABA and salinity stress but enhanced stress tolerance by promoting centromere-binding factor 3 and DREB2A expression [34]. PKE1 functions similarly with TaCHP. *PKE1* was downregulated by ABA and drought treatment (Figure 4). Salt tolerance in transgenic *PKE1* OE plants significantly increased, compared with that of WT plants (Figure 5 and Figure 6). The molecular mechanisms, regarding how the OE of negative regulators enhances salt tolerance, remain to be elucidated. 

PKE1 can bind to F-box proteins [35].This is a clue for understanding the molecular mechanisms of the negative regulator PKE1 in enhancing salt tolerance in tobacco. F-box proteins are involved in post-transcriptional regulation by targeted protein ubiquitination [36]. F-box protein gene plays an important role in abiotic stress [37,38,39,40] and plant miRNA function in plant [41]. For example, an F-box protein DOR inhibits the ABA-induced stomatal closure under drought stress in *Arabidopsis* [39], and another F-box protein MAX plays an important role in the regulation of plant growth and development and in response to abiotic stress conditions [38]. The high or constant expression of *PKE1* in reproductive tissues implies its special function in salt tolerance at the reproductive stage (Figure 3A,B). The result coincides with the hypermethylation of *PKE1* DNA sequences in leaf rather than that during fruit development (Figure 3C), because DNA methylation inhibits gene expression [42]. DNA methylation plays a crucial role in plant organs and genotypes specific for regulating gene expression responsive to environmental stress, and it is a highly important regulatory mechanism for plant adaptation to environmental stresses [43,44,45,46,47]. This condition leads to the problem of identifying the cause of PKE1 hypermethylation in leaf but not in reproductive organs and implies that PKE1 confers tolerance to salt stress not only in the seedlings but also in the reproductive stage. These findings, together with the result that PKE1 can bind to F-box proteins, show that the *PKE1* modulates the salt tolerance involved in post-transcriptional regulation.

## 4. Materials and Methods

### 4.1. Plant Materials and Growth Conditions

Tomato plants (*S. pennellii* LA0716 and *S. lycopersicum* cv. M82) were grown in a greenhouse under a 16 h light/8 h dark regime at approximately 25 °C. At the six-leaf stage the seedlings were treated with drought, methyl viologen (MV) stress, and different hormones for *PKE1* stress-responsive expression analysis. Under drought dehydration (DH) stress, the seedlings, removed from the soil, were placed on a filter paper. For MV and hormone treatments, the seedlings were sprayed with solutions containing 100 µM MV, 100 µM ABA, 100 µM of gibberellic acid (GA_3_), 100 µM SA, 100 µM ethephon (Eth; an ethylene releaser), or distilled water (control). *PKE1* expression was validated in the different organs of *S. pennellii* and *S. lycopersicum* cv. M82 at the six-leaf stage of seedlings. Then leaves were collected at designated time points, and different tissues were immediately frozen in liquid nitrogen and stored at −80 °C until use. 

### 4.2. Gene Isolation, Vector Construction, and Genetic Transformation

Tomato *PKE1* (*SpPKE1*: Sopen05g032700; *SlPKE1*: Solyc05g054210, http://solgenomics.net/, accessed on: 18 May 2019) was isolated from *S. pennellii* and *S. lycopersicum* cv. M82, respectively, the vector construction and genetic transformation were described as our previous study [35]. Briefly, the 35S-*SpPKE1* and RNAi plasmid was transformed into tomato cultivar M82 and tobacco (*Nicotiana nudicaulis*) by *Agrobacterium tumefaciens* (strain C58)-mediated transformation. After screening the regenerated shoots on the selection medium containing kanamycin, the transgenic plants were further verified via PCR with genomic DNA as template using 35S promoter forward and gene-specific reverse primers (Appendix A).

### 4.3. Bioinformatics Analysis

Exons and introns were identified by comparing the genomic DNA and cDNA sequences by using Gene Structure Display Server (http://gsds.cbi.pku.edu.cn, accessed on: 18 May 2019) [48]. Homologous proteins of PKE1 were collected from the EBI database (http:// www.ebi.ac.uk/, accessed on: 18 May 2019) using BLASTP. A phylogenetic tree was constructed using the neighbor-joining (NJ) method with MEGA (version 5.05) software [49]. The DNA methylation level of PKE1 was analyzed based on the tomato epigenome database (http://ted.bti.cornell.edu/epigenome/, accessed on: 18 May 2019) [50] by comparing the cytosine (C) and methylcytosine (5mC) of the genomes; the cytosine methylation ratio = 5mC/(5mC+C). The *PKE1* promoter sequences were isolated from BLASTN search by using the *PKE1* gene sequence queries against the tomato whole genome scaffolds (version 2.40) and *S. pennellii* WGS chromosome data at the SGN website (https://solgenomics.net/, accessed on: 18 May 2019). The promoter sequences (1.5 kb upstream of 5′ UTR) of *SpPKE1* and *SlPKE1* were submitted to the PlantCARE database (http://bioinformatics.psb.ugent.be/webtools/plantcare/html/, accessed on: 18 May 2019) for cis-element prediction.

### 4.4. Subcellular Localization of SpPKE1

*SpPKE1* coding region (without stop codon) was amplified by PCR from the *S. pennellii* LA0716 cDNA with primers containing *Kpn*I and *BamH*I restriction sites (Appendix A) for subcellular localization analysis. The PCR product was cloned into the pMD18-T vector (TaKaRa, Shiga, Japan), and sequenced. The correct sequence of *SpPKE1* in plasmid was digested with *Kpn*I and *BamH*I, and the fragment was fused to the 35S promoter with fusion construction of enhanced green fluorescent protein (EGFP) into the *Kpn*I and *BamH*I-digested pCAMBIA1391 vector [51]. The resulting *35S:SpPKE1*-*EGFP* fusion construct with the GFP alone (*35S-EGFP*) was bombarded into BY-2 (*N. tabacum* cv. Bright Yellow 2) tobacco cells by using Biolistic PDS-1000 (Bio-Rad, Hercules, CA, USA). All samples were observed under a Leica TCSST2 confocal laser microscope (Zeiss, LSM510, Oberkochen, Germany) after 24 h of bombardment.

### 4.5. RNA Isolation and Quantitative Reverse Transcription-PCR (qRT-PCR)

Gene expression patterns were examined by isolating total RNA by using TRIzol solution (Sangon, Shanghai, China). DNase I-treated total RNA was used for first-strand cDNA synthesis, which adopted a PrimeScript RT reagent Kit (TaKaRa, Dalian, China) according to the manufacturer’s instructions. The resulting cDNA was used for RT-PCR, which was performed on CFX96 (Bio-Rad, USA) with Eva Green SMX (Bio-Rad) by employing primers specific for genes as shown in Appendix A. PCR amplification consisted of an initial incubation at 94 °C for 5 min, followed by 40 cycles of 94 °C for 10 s, 58 °C for 15 s, and 72 °C for 20 s. Data was gathered during the extension step. Melting-curve acquisition and analysis was performed on cycler. Each sample included three replicates, and each assayed sample represented three independently collected samples. Data was normalized against the reference β-actin gene (Solyc11g005330.1.1). 

### 4.6. Salt Tolerance Testing of Transgenic Plants

After anti-kanamycin analysis [52] and PCR (35S promoter forward and gene-specific reverse primers) confirmation, the resulted uniform-sized T_3_ homozygous at three-leaf stage tomato seedlings [35] were transplanted into cylindrical pots (Diameter: 8 cm, Height: 8 cm) and nourished to grow until the four-leaf stage for salt treatment, in order to evaluate salt tolerance of transgenic tomato lines. Salt stress was initiated by watering 200 mM NaCl. After 14d treatment, photos were taken and the survival rate was evaluated.

In order to simulate salt treatment, T_2_ transgenic tobacco [35] were selected by germinating seeds on 1/2 MS medium containing 50 mg/L kanamycin. After germination, the germinant positive seedlings together with WT were transplanted in 1/2 MS medium with 200 mM NaCl and grew for 12 days, the test was repeated 3 times. 12 days after salt treatment, the root length, chlorophyll, and malondialdehyde (MDA) content were measured. After removing the seedling from the medium, we used a ruler to measure the root length of each seedling. Chlorophyll content was measured using the Lichtenthaler’s method [53]. Leaf tissues were ground under liquid nitrogen and extracted with 8 mL of 95% (*v*/*v*) ethyl alcohol. Absorption spectra were detected at 665 and 649 nm. Chlorophyll was computed using the following equation: chlorophyll concentration (mg/mL) = (6.63 × A665) + (18.08 × A649), where *A* refers to the absorbance at a specified wavelength. MDA was assayed for indirect evaluation of lipid peroxidation by using thiobarbituric acid as described previously [54]. 

### 4.7. Statistical Analysis

The results of salt tolerance testing and qRT-PCR experiments were displayed as mean ± standard error (SE). Data was analyzed using variance by SAS software (version 8.0, SAS Institute, Cary, NC, USA), and statistical differences were compared using Fisher’s least significant difference (LSD) test.

## 5. Conclusions

Wild species often show more tolerance to abiotic stress than their cultivated counterparts. Genes from wild species is the best resource of improving abiotic resistance in cultivated species. In this study, a tomato proline-, lysine-, and glutamic-rich type gene *SpPKE1* was isolated from drought-resistant species (*S. pennellii* LA0716) enhanced the salt tolerance in cultivated tomato and tobacco. Moreover, expression and methylation analysis results indicated that the *PKE1* involved in post-transcriptional regulation.

## Figures and Tables

**Figure 1 ijms-20-02478-f001:**
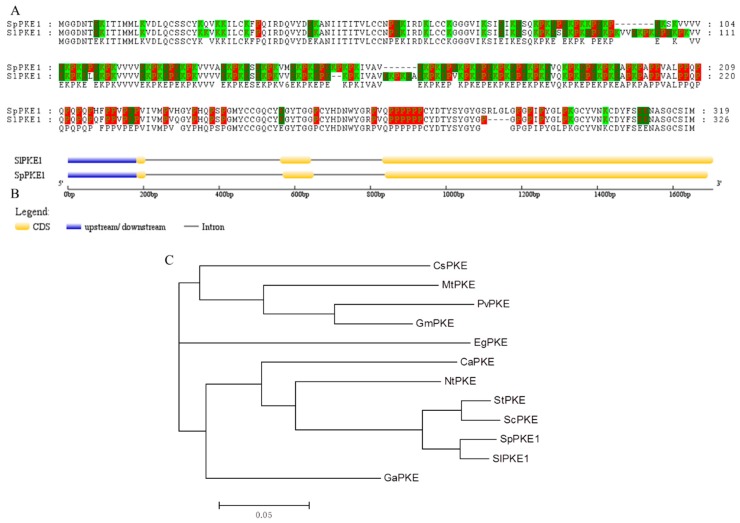
Comparison of amino acid sequence and phylogenetic analyses of PKE1. (**A**) Amino acid alignment of *S. lycopersicum cv.* M82 (SlPKE1) and *S. pennellii* LA0716 (SpPKE1). (**B**) Gene structure of the tomato *SlPKE1* and *SpPKE1* generated from Gene Structure Display Server (GSDS). The yellow block indicates the coding sequence (CDS), the blue block refers to upstream or downstream of the genes, the black line represents the intron. Scale bar indicates the DNA sequence length. (**C**) Neighbor Joining tree for SlPKE1, SpPKE1 and PKE1 from their highest similarity proteins in the EBI (European Bioinformatics Institute) database. The PKE1 from other species are as follows: CsPKE1 (*Citrus sinensis*: A0A067GT86), MtPKE1 (*Medicago truncatula1*: G7JMW9)*,* PvPKE1 (*Phaseolus vulgaris*: V7CAV2*)*, GmPKE1 (*Glycine max*: I1MQY4), EgPKE1 (*Erythranthe guttata*: A0A022QI88), CaPKE1 (*Capsicum annuum*: Accession no. O81922), NtPKE1 (*Nicotiana tabacum*: A0A1S4A831), StPKE1 (*Solanum tuberosum*: M1AFN1), ScPKE1 (*Solanum chacoense*: A0A0V0I462) and GaPKE1 (*Gossypium arboretum*: A0A0B0MHK6).

**Figure 2 ijms-20-02478-f002:**
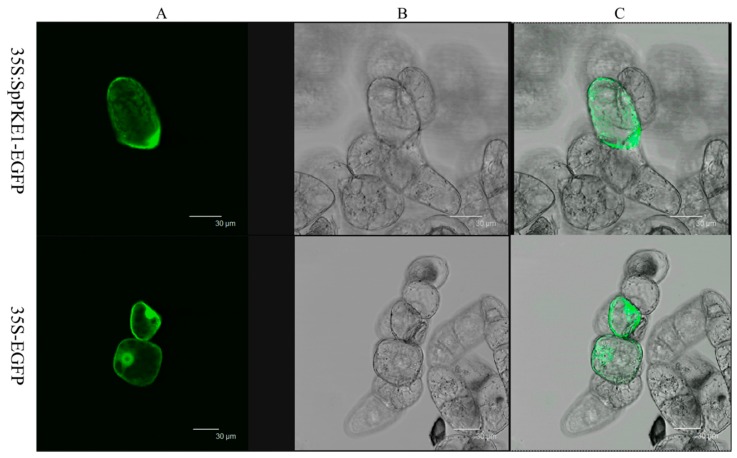
Subcellular localization of SpPKE1 in cv BY-2 tobacco cells. CaMV 35S::*EGFP* and 35S::*SpPKE1*-*EGFP* constructs were expressed transiently in cv BY-2 tobacco cells. Green fluorescent protein (GFP) fluorescence images (**A**), Bright-field images (**B**), and merged images (**C**) of representative cells transformed with 35S::*EGFP* or 35S::*SpPKE1*-*EGFP* fusion protein.

**Figure 3 ijms-20-02478-f003:**
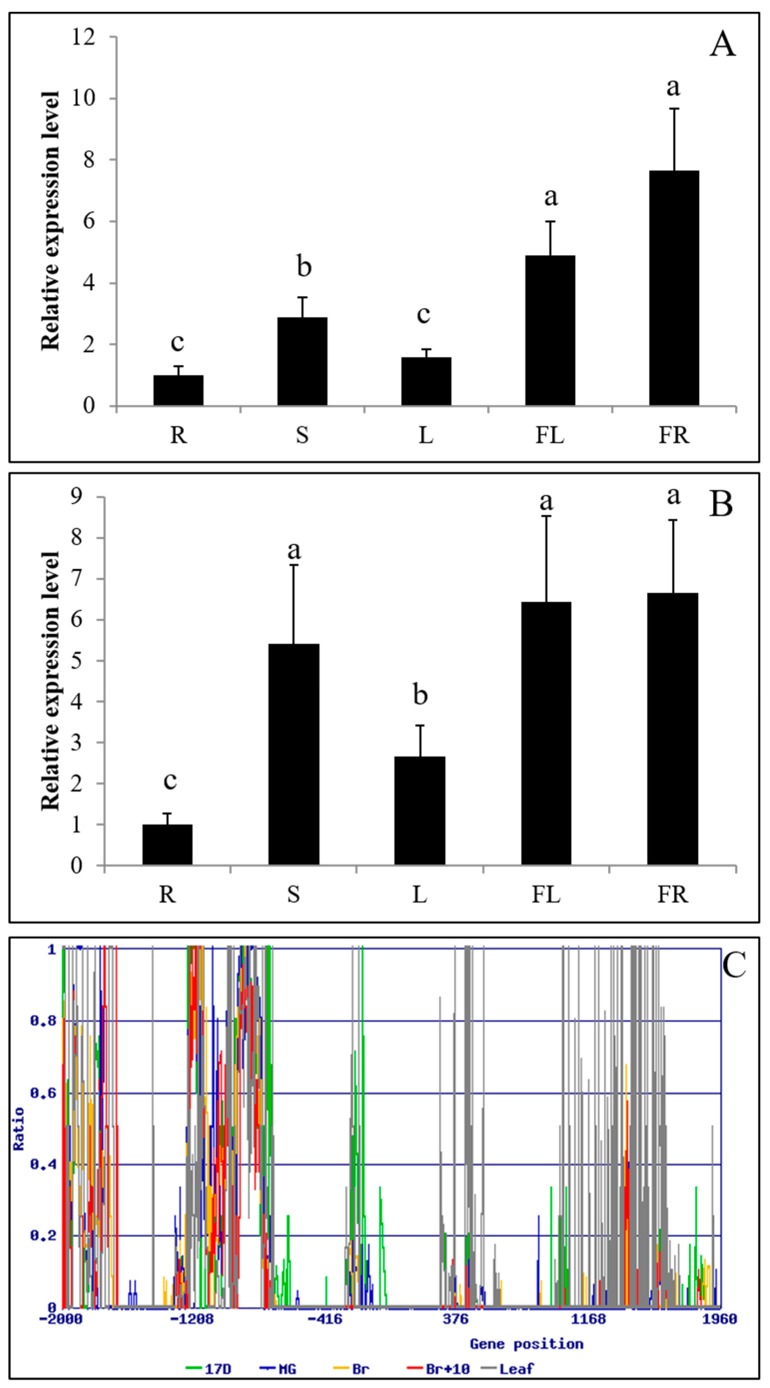
Tissue expression and methylation analysis of *PKE1*: gene expression profile in different tissues (R: root, S: stem, L: leaf, FL: flower, FR: fruit) of *S. pennellii* LA0716 (**A**) and *S. lycopersicum* cv. M82 (**B**). All samples were collected at the indicated time points from three biological replicates for each treatment condition. Error bars indicate ± SE of means at *p* < 0.05 (*n* = 3). (**C**) Analysis of the DNA methylation of *PKE1* promoter and coding sequences at different tissues and stages. Immature (17 DPA), mature green (MG: 39 DPA), breaker (Br: 42 DPA), and red ripe (Br+10: 52 DPA), and leaf. Ratio = 5mC/(5mC+C).

**Figure 4 ijms-20-02478-f004:**
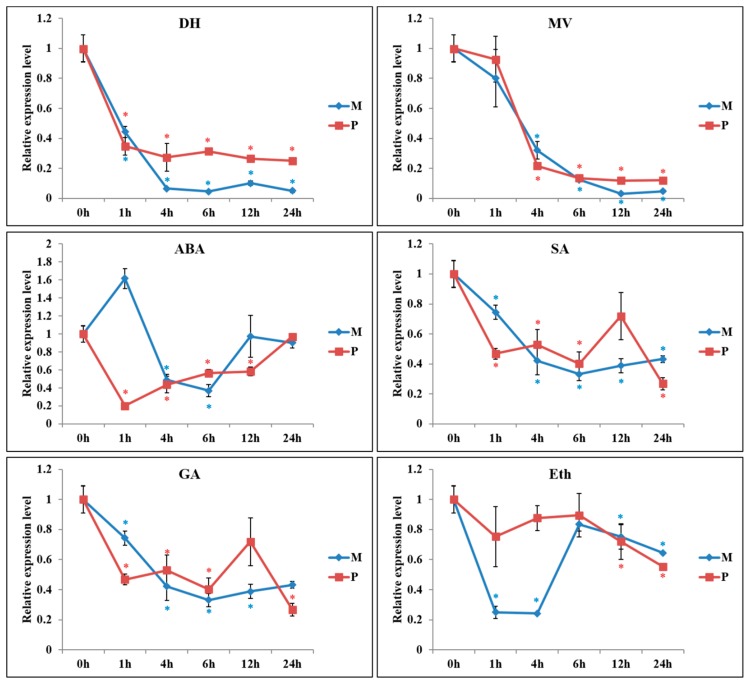
Expression pattern of tomato *PKE1* under drought dehydration (DH), methyl viologen (MV), ABA (Abscisic acid), gibberellic acid (GA_3_), SA (salicylic acid), ethephon (Eth; an ethylene releaser) treatment in *S. lycopersicum* cv. M82 (M) and *Solanum pennellii* LA0716 (P). All samples were collected at the time points (‘h’ refers to hours) after treatment from three biological replicates of each treatment. Data shown are means ± SE (*n* = 3) *, Means differ *p* < 0.05.

**Figure 5 ijms-20-02478-f005:**
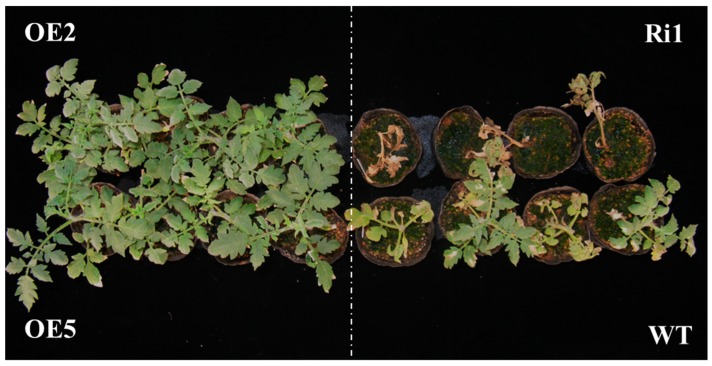
Seedling phenotype of tomato with overexpressed *PKE1* (OE) and RNAi knockdown (Ri) under salt stress conditions. Four-leaf stage seedlings for salt treatment initiated by watering 200 mM NaCl for 14 d.

**Figure 6 ijms-20-02478-f006:**
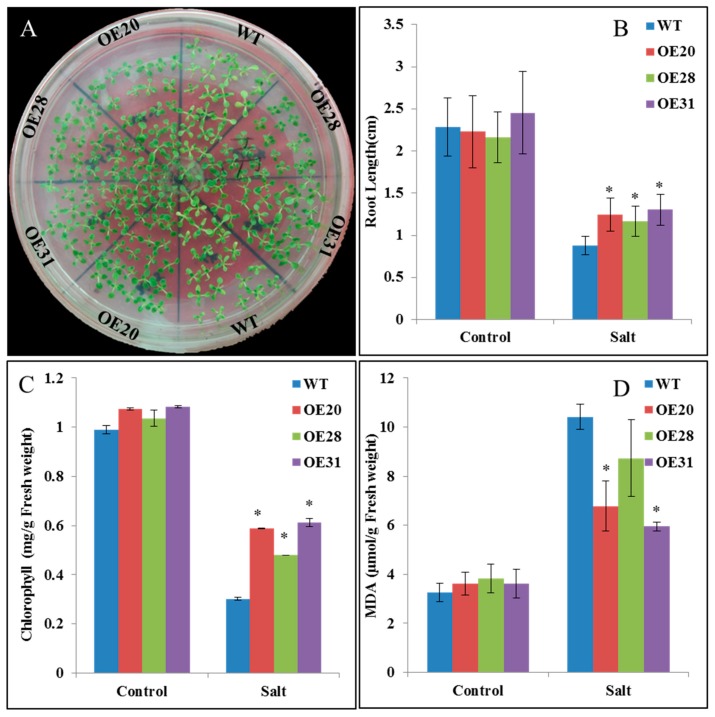
The overexpression (OE) of *PKE1* in tobacco enhanced its salt tolerance. (**A**) OE of tobacco *SpPKE1* lines and wild type (WT) subjected to 200 mM NaCl. The image was captured after 7 days of treatment. The root length (**B**), contents of chlorophyll (**C**), and malondialdehyde (MDA) (**D**) changes subjected to 200 mM NaCl (Salt) and absence of NaCl (Control) in transgenic and WT plants after 12 days of treatment. All data presented as means ± SE, *, Means differ *p* < 0.05.

**Table 1 ijms-20-02478-t001:** The *cis*-elements identified in the promoters of *SlPKE1* (in M82) and *SpPKE1* (in *S. pennellii*) and the differences of their *cis*-elements.

*S. pennellii*	M82	Functions of *cis*-elements
5UTR Py-rich stretch	5UTR Py-rich stretch	
AAGAA-motif	AAGAA-motif	
AC-II	AC-I	
AE-box	AE-box	
	★ARE	cis-acting regulatory element essential for the anaerobic induction
AT-rich element	AT-rich element	
	★AT1-motif	part of a light responsive module
ATCT-motif	ATCT-motif	
	★ATGCAAAT motif	cis-acting regulatory element associated to the TGAGTCA motif
Box 4	Box 4	
Box I	Box I	
Box-W1	Box-W1	
CAAT-box	CAAT-box	
	★CGTCA-motif	cis-acting regulatory element involved in the MeJA-responsiveness
	★GA-motif	part of a light responsive element
GAG-motif	GAG-motif	
GT1-motif	GT1-motif	
HSE	HSE	
☆LTR		cis-acting element involved in low-temperature responsiveness
	★I-box	part of a light responsive element
O2-site	O2-site	
Skn-1_motif	Skn-1_motif	
Sp1	Sp1	
TATA-box	TATA-box	
TC-rich repeats	TC-rich repeats	
TCA-element	TCA-element	
TCT-motif	TCT-motif	
	★TGACG-motif	cis-acting regulatory element involved in the MeJA-responsiveness
W box	W box	
circadian	circadian	

☆ (★) symbol means the *cis*-elements appear in the PKE1 promoters of *S. pennellii* (M82), but not in M82 (*S. pennellii*).

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
