# Peer review of "SpPKE1,* a Multiple Stress-Responsive Gene Confers Salt Tolerance in Tomato and Tobacco"

_ijms, 2019, doi:10.3390/ijms20102478_

Round 1

Reviewer 1 Report

Authors successfully addressed most of my comments. I would like to recommend to accept this manuscript for publication in IJMS.

One more minor comment:

Ln146-148 should be revised from "Expression profile of SpPKE1 (A) and S. lycopersicum cv. M82 (B) in the different tissues (R: root, S: stem, L: leaf, FL: flower, FR: fruit) of S. pennellii LA0716." to "Expression profile of PKE1 in the different tissues (R: root, S: stem, L: leaf, FL: flower, FR: fruit) of S. pennellii LA0716 (A) and S. lycopersicum cv. M82 (B)."

Author Response

Thanks for your suggestion. We have revised it as suggested.

Much appreciated for your thorough review and for the helpful comments regarding our manuscript

Reviewer 2 Report

2019.5.2

Reviewer #2 response to Authors: I was surprised and astonished that the Authors removed their previous BBRC paper from the list of references. In my opinion this is a confirmation of my suspect about the choice of the Authors that would like to obtain two different manuscripts starting from very similar data and this is not acceptable for an international journal like IJMS. In brief, I am reporting above a comparison of title and Authorship between the already published manuscript and that submitted to IJMS:   

Jinhua Li, Yaling Wang, Juanjuan Wei, Yu Pan, Chenggang Su, Xingguo Zhang (2018) A tomato proline-, lysine-, and glutamic-rich type gene SpPKE1 positively regulates drought stress tolerance. Biochemical and Biophysical Research Communications 499, (4), 777‐782.

Jinhua Li, Chunrui Chen, Juanjuan Wei, Yu Pan, Chenggang Su, Xingguo Zhang. SpPKE1, a multiple stress-responsive proline-, lysine-, and glutamic-rich type gene confers salt stress tolerance in tomato and tobacco. IJMS (submitted).

I would invite the Authors and the Editor to verify how many details between the manuscripts are shared. I really understand the differences between drought and salt stress, but I suggested the Authors to refer in the new manuscript to the previous ones avoiding to repeat some methods and others. Instead of the Authors have changed the reference list deleting their previous paper and this is in my opinion unacceptable. Based on this consideration I suggest to reject the manuscript “SpPKE1, a multiple stress-responsive proline-, lysine-, and glutamic-rich type gene confers salt stress tolerance in tomato and tobacco” by Li et al.

Updated comments (2019.5.5):

I read the further clarification letter of the Authors of the manuscript “SpPKE1, a multiple stress-responsive proline-, lysine-2 , and glutamic-rich type gene confers salt stress 3 tolerance in tomato and tobacco” by Li et al. First of all, I apologize for my mistake in the second round reviewing the manuscript because of I was not able to find in the new list of references the #35 that reported previous paper by Li et al on BBRC.

Based on this my preliminary mistake, I loss to read carefully the other potential limits and doubt that the manuscript solicited me previously. Thus, today I have red again the manuscript that is acceptable for publication after major revisions and not in this present form.

My first request is to really shorten the manuscript title as suggested also by other reviewers and I could suggest “SpPKE1, a multiple stress-responsive gene confers salt tolerance in tomato and tobacco”. Then, I retain the English style of whole manuscript very poor, mainly Results and M&M sections. 

I also suggest the Authors to find a better reference to replace the reference [1], there are many other indexed papers related to the same topic instead of the review indicated by the Authors.

I found M&M section not enough detailed in different parts; i.e. in section 2.6 the Authors referred to “uniform-sized T3 homozygous three-leaf stage seedlings” please explain homozygous for what, and how the selection was performed.

What does it mean “For simulating salt treatment, positive T2 transgenic tobacco seedlings were germinated in 1/2 MS medium for seven days and then sub-cultured in 1/2 MS containing 200 mM NaCl”? Further did the Authors report the methods for testing the salt tolerance of transgenic plants in both tomato and tobacco? It is not clear and well described.

The Authors wrote about root length without describing any method for measuring them, please add in the description method. 

Please refer to your BBRC previous paper [35] also in the subchapter 4.6 of M&M section.

Please rewrite more clearly Figure 2 legend and/or treatment acronyms. There is a discrepancy between the description and the acronym indicated in the figure. It is not automatic understanding that the treatment “PR” means pennellii root.

I request the Authors to perform/report the statistical analysis on RT-PCR data.

I am very surprised that the Authors do not use among abiotic stress the salt stress; could they explain why? In addition, I suggest substantial modifications also in section 2.3 starting from the title that could be “2.3. PKE1 expression suppression by abiotic stress and hypermethylation in different tissues”.

Furthermore, the whole subsection 2.3 could be re-written as follow “RT-PCR analysis showed similarity of PKE1 tissue expression between S. pennellii and S. lycopersicum cv. M82. Indeed, PKE1 was highly expressed in flower and fruit, while a lower expression was observed in root and leaf (Figures 3A and 3B). DNA methylation analysis indicated that PKE1 promoter and the coding sequences were highly methylated, and accordingly with gene expression, methylation was significantly higher in the leaf compared to flower and fruit (Figures 3C and S1). Interestingly, PKE1 expression was significantly suppressed by various abiotic stresses, including drought, MV, ABA, and SA (Figure 4). Under drought, MV, GA3, and SA treatments, PKE1 expression was gradually reduced, and S. pennellii and S. lycopersicum cv. M82 showed similar expression patterns. However, PKE1 expression returned to pre-treatment levels after 24 h from ABA and Eth treatment. Under ABA treatment, PKE1 expression was significantly suppressed after 1 h of treatment in S. pennellii, while  none variation in gene expression was observed in S. lycopersicum cv. M82. By contrast, under Eth treatment, PKE1 expression was significantly suppressed after 1 h of treatment in tomato cv. M82 but did not exhibit response in S. pennellii”.

The last sentence “Hence, PKE1 is a negative regulator of abiotic stress and manifests roughly similar expression patterns in S. pennellii and S. lycopersicum cv. M82” could be deleted or better re-written for discussion section.

Also the last subsection (2.4) of Results must be completely revised for English style but not only. First of all the title that could be as follow “2.4. PKE1 overexpression enhanced tolerance to salt stress”. In addition the Authors must revise the subsection for English style that is very poor and explain the sentence “After drought treatment, less leaf necrosis and increased shoot biomass were observed in the transgenic OE plant lines”; did they refer to salt treatment? Further there are many mistakes as “overexporession” and “RNAi-konckdown” only in the first row. Instead of “enhanced salt resistance” it is better to write salt tolerance.

In section 2.2 please modify the title in “SpPKE1 localizes into the cytoplasm” and delete the last short sentence that is obvious.

I warmly suggest the Authors to better explain the sentence related to Figure S1 “In previous studies on drought stress in tomato introgression lines[15], a differential expression profile of the PKE1 gene was observed between the drought-tolerant introgression line and M82 (Fig. S1)”,

being a supplemental material but very important for its content the sentence could be rewritten.

I found the discussion subsection title too long, please, shorten them in the revision form. I suggest for the first one the follow “PKE1 gene sequence diversity between Solanum species and involvement in abiotic stress”. The second ones could be changed in “PKE1 abiotic stress down-regulation and salt tolerance”.

Finally, I kindly ask the Authors to prepare a new revised version of their manuscript taking into account all the requests/suggestions, having care to use the tracked revision tool of word so that I can attention all their modifications and the general improvement of the manuscript. Taking care to explain well because they did not include the salt stress in the abiotic stress applied to wild type plants/species before using salt stress treatment after tomato and tobacco genetic transformation.

Waiting for a revised version of the manuscript and apologizing again for previous review I remain

Sincerely Yours

Author Response

My first request is to really shorten the manuscript title as suggested also by other reviewers and I could suggest “SpPKE1, a multiple stress-responsive gene confers salt tolerance in tomato and tobacco”. Then, I retain the English style of whole manuscript very poor, mainly Results and M&M sections.

Response(R): Thanks for your suggestion. The title has been revised as suggested. We have fully revised the manuscript based on the comments, and asked a native English speaker to improve the manuscript. Changes made to the text are "Track Changes" so that you can be easily identified.

I also suggest the Authors to find a better reference to replace the reference [1], there are many other indexed papers related to the same topic instead of the review indicated by the Authors.

R: This has been revised according to the suggestions.

I found M&M section not enough detailed in different parts; i.e. in section 2.6 the Authors referred to “uniform-sized T3 homozygous three-leaf stage seedlings” please explain homozygous for what, and how the selection was performed.

R: Thanks for your suggestion. We have added more details as below in M&M section to explain homozygous for what, and how the selection was performed. At 2 or 3 leaf stage, by spray kanamycin (100mg/L, once a day for three days), after 1 week, the non-transgenic seedling leaves began to have yellow spots but no yellow spots in transgenic seedling leaves. If the all the plant shows anti-kanamycin that no yellow spots appeared, then further verified via PCR with genomic DNA as template using 35S promoter forward and gene-specific reverse primers. It suggests that the seedling is homozygous. And it is added in 4.6. Changes made to the text are on lines 645 to 646.

What does it mean “For simulating salt treatment, positive T2 transgenic tobacco seedlings were germinated in 1/2 MS medium for seven days and then sub-cultured in 1/2 MS containing 200 mM NaCl”? Further did the Authors report the methods for testing the salt tolerance of transgenic plants in both tomato and tobacco? It is not clear and well described.

R: Thanks for your question. We have corrected it accordingly as below.

For simulating salt treatment, positive transgenic plants of T2 transgenic tobacco seedlings were selected by germinating seeds on 1/2 MS medium containing 50 mg/L kanamycin. After germination, the positive seedlings were transplanted in 1/2 MS medium containing 200 mM NaCl and grown for 12 days. Besides, we have added more details in M&M section 4.6. Changes made to the text are on lines 647 to 656.

The Authors wrote about root length without describing any method for measuring them, please add in the description method.

R: Thanks for your question. We have added it accordingly. Changes made to the text are on lines 655 to 656.

Please refer to your BBRC previous paper [35] also in the subchapter 4.6 of M&M section.

R: Thanks for your question. We have added it accordingly. Changes made to the text are on lines 647 and 652.

Please rewrite more clearly Figure 2 legend and/or treatment acronyms. There is a discrepancy between the description and the acronym indicated in the figure. It is not automatic understanding that the treatment “PR” means pennellii root.

R: Thanks for your suggestion. We have revised it accordingly.

I request the Authors to perform/report the statistical analysis on RT-PCR data.

R: Thanks for your suggestion. We have added statistical analysis on RT-PCR data, and the statistical analysis was added in the figures 3 A&B and figure 4, the statistical analysis method was added in M&M section 4.7.

I am very surprised that the Authors do not use among abiotic stress the salt stress; could they explain why? In addition, I suggest substantial modifications also in section 2.3 starting from the title that could be “2.3. PKE1 expression suppression by abiotic stress and hypermethylation in different tissues”.

R: Thanks for your suggestion. We have revised it accordingly.

Actually, we have performed the expression analysis twice and the results are consistent. The representative result of the experiment is presented in the manuscript. In the first time, we only analyzed the expression in S. pennellii of PKE1 respond to drought, ABA, Eth, GA, MV, wounding, cold, heat and salt treatment. Please see the attached supporting figure 1 (the expression of PKE1 response to salt stress in S. pennellii). It shows that PKE1 was also suppressed by salt stress. In the second experiment, we compared the expression of PKE1 in S. pennellii and S. lycopersicum cv. M82. The result of this experiment is presented in the manuscript.

Furthermore, the whole subsection 2.3 could be re-written as follow “RT-PCR analysis showed similarity of PKE1 tissue expression between S. pennellii and S. lycopersicum cv. M82. Indeed, PKE1 was highly expressed in flower and fruit, while a lower expression was observed in root and leaf (Figures 3A and 3B). DNA methylation analysis indicated that PKE1 promoter and the coding sequences were highly methylated, and accordingly with gene expression, methylation was significantly higher in the leaf compared to flower and fruit (Figures 3C and S1). Interestingly, PKE1 expression was significantly suppressed by various abiotic stresses, including drought, MV, ABA, and SA (Figure 4). Under drought, MV, GA3, and SA treatments, PKE1 expression was gradually reduced, and S. pennellii and S. lycopersicum cv. M82 showed similar expression patterns. However, PKE1 expression returned to pre-treatment levels after 24 h from ABA and Eth treatment. Under ABA treatment, PKE1 expression was significantly suppressed after 1 h of treatment in S. pennellii, while  none variation in gene expression was observed in S. lycopersicum cv. M82. By contrast, under Eth treatment, PKE1 expression was significantly suppressed after 1 h of treatment in tomato cv. M82 but did not exhibit response in S. pennellii”.

The last sentence “Hence, PKE1 is a negative regulator of abiotic stress and manifests roughly similar expression patterns in S. pennellii and S. lycopersicum cv. M82” could be deleted or better re-written for discussion section.

R: Thanks for your excellent suggestion. We have revised it accordingly, and the last sentence was deleted.

Also the last subsection (2.4) of Results must be completely revised for English style but not only. First of all the title that could be as follow “2.4. PKE1 overexpression enhanced tolerance to salt stress”. In addition the Authors must revise the subsection for English style that is very poor and explain the sentence “After drought treatment, less leaf necrosis and increased shoot biomass were observed in the transgenic OE plant lines”; did they refer to salt treatment? Further there are many mistakes as “overexporession” and “RNAi-konckdown” only in the first row. Instead of “enhanced salt resistance” it is better to write salt tolerance.

R: It refers to salt treatment. Sorry for the mistake, we have corrected and revised the manuscript thoroughly.

In section 2.2 please modify the title in “SpPKE1 localizes into the cytoplasm” and delete the last short sentence that is obvious.

R: Thanks for your suggestion. We have modified the title and deleted the last short sentence accordingly.

I warmly suggest the Authors to better explain the sentence related to Figure S1 “In previous studies on drought stress in tomato introgression lines[15], a differential expression profile of the PKE1 gene was observed between the drought-tolerant introgression line and M82 (Fig. S1)”, being a supplemental material but very important for its content the sentence could be rewritten.

R: Thanks for your kind suggestion. We have added the details in the result. The Figure S1 is from the published data after we analyzed, and we want to explain how the gene is observed, and its expression responsive to drought stress in S. pennellii and S. lycopersicum cv. M82 were further RT-PCR confirmed, thus we put it in supplemental material.

I found the discussion subsection title too long, please, shorten them in the revision form. I suggest for the first one the follow “PKE1 gene sequence diversity between Solanum species and involvement in abiotic stress”. The second ones could be changed in “PKE1 abiotic stress down-regulation and salt tolerance”.

R: Thanks for your suggestion. We have revised it accordingly.

Round 2

Reviewer 2 Report

The last version of the manuscript by Li et al. contains rather all the suggestions and request of modifications the text. I have two last minor requests for the Authors. First, I suggest to rewrite the legend of Figure 3 as follow "Tissue expression and methylation analysis of PKE1: gene expression profile in different tissues (R: root, S: stem, L: leaf, FL: flower, FR: fruit) of (P) S. pennellii LA0716 (A) and (M) S. lycopersicum cv. M82 (B)......" As alternative, as my previous suggestion you could modify the figures 3A and 3B deleting P and M by all the treatments and the legend could become "Tissue expression and methylation analysis of PKE1: gene expression profile in different tissues (R: root, S: stem, L: leaf, FL: flower, FR: fruit) of (P) S. pennellii LA0716 (A) and (M) S. lycopersicum cv. M82 (B)......" 

The second suggestion is related to section 4.6. The Authors refer to a spray treatment with kanamycin to select transgenic plants that is correct but I suggest to cite an old reference (Sunseri F, Fiore MC, Mastrovito F, Tramontano E, Rotino GL (1993) In vivo selection and genetic analysis for kanamycin resistance in transgenic eggplant (Solanum melongena L.). J. Genet. & Breed. 47: 299-306 2-s2.0-0027131172) that demonstrated the closed correlation between leaf bleching after kanamycin spraying and the absence of the transgene and viceversa, so that you can not refer to a PCR analysis that became redundant. Finally, I warmly suggest the Authors to rewrite the sentence on lines 300-302 in "Positive T2 transgenic tobacco seedlings [35]were selected by germinating seeds on 1/2 MS medium containing 50 mg/L kanamycin. After germination, the positive seedlings were transplanted in 1/2 MS medium containing 200 mM NaCl to simulate salt treatment and grown for 12 days, each with three replicates".

After this last minor modifications, I retain the manuscript suitable for publication on IJMS.

Author Response

The last version of the manuscript by Li et al. contains rather all the suggestions and request of modifications the text. I have two last minor requests for the Authors. First, I suggest to rewrite the legend of Figure 3 as follow "Tissue expression and methylation analysis of PKE1: gene expression profile in different tissues (R: root, S: stem, L: leaf, FL: flower, FR: fruit) of (P) S. pennellii LA0716 (A) and (M) S. lycopersicum cv. M82 (B)......" As alternative, as my previous suggestion you could modify the figures 3A and 3B deleting P and M by all the treatments and the legend could become "Tissue expression and methylation analysis of PKE1: gene expression profile in different tissues (R: root, S: stem, L: leaf, FL: flower, FR: fruit) of (P) S. pennellii LA0716 (A) and (M) S. lycopersicum cv. M82 (B)......" 

Response (R)Thanks for your suggestion. The Figure 3 and legend has been revised as suggested.

The second suggestion is related to section 4.6. The Authors refer to a spray treatment with kanamycin to select transgenic plants that is correct but I suggest to cite an old reference (Sunseri F, Fiore MC, Mastrovito F, Tramontano E, Rotino GL (1993) In vivo selection and genetic analysis for kanamycin resistance in transgenic eggplant (Solanum melongena L.). J. Genet. & Breed. 47: 299-306 2-s2.0-0027131172) that demonstrated the closed correlation between leaf bleching after kanamycin spraying and the absence of the transgene and viceversa, so that you can not refer to a PCR analysis that became redundant. Finally, I warmly suggest the Authors to rewrite the sentence on lines 300-302 in "Positive T2 transgenic tobacco seedlings [35]were selected by germinating seeds on 1/2 MS medium containing 50 mg/L kanamycin. After germination, the positive seedlings were transplanted in 1/2 MS medium containing 200 mM NaCl to simulate salt treatment and grown for 12 days, each with three replicates".

R: Thanks for your suggestion. We have cited the reference in [52]. In our experiment, after kanamycin spraying (verify NPT II gene), we still PCR (35s +Rv) confirmation the PKE1 gene, thus we keep the PCR analysis in the M&M sections. 300-302 we have revised it. Changes made to the text are on lines 332 to 335